# Characterization of the endometrial, cervicovaginal and anorectal microbiota in post-menopausal women with endometrioid and serous endometrial cancers

Gregory M. Gressel[1,2]*, Mykhaylo Usyk[3], Marina Frimer[4], D. Y. S. Kuo[1,2], Robert D. Burk[1,2,3,5,6]

**1** Division of Gynecologic Oncology, Department of Obstetrics & Gynecology and Women's Health, Albert Einstein College of Medicine, Montefiore Medical Center, Bronx, NY, United States of America, **2** Albert Einstein Cancer Center, Albert Einstein College of Medicine, Bronx, NY, United States of America, **3** Department of Pediatrics, Albert Einstein College of Medicine, Montefiore Medical Center, Bronx, NY, United States of America, **4** Department of Obstetrics & Gynecology, Karches Center for Oncology Research, Feinstein Institutes at Northwell Health, Zucker School of Medicine at Hofstra/Northwell, Hempstead, NY, United States of America, **5** Department of Microbiology & Immunology, Albert Einstein College of Medicine, Bronx, NY, United States of America, **6** Department of Epidemiology & Population Health, Albert Einstein College of Medicine, Bronx, NY, United States of America

* gregory.gressel@spectrumhealth.org

## Abstract

### Objective

To characterize the microbiota of postmenopausal women undergoing hysterectomy for endometrioid (EAC) or uterine serous cancers (USC) compared to controls with non-malignant conditions.

### Methods

Endometrial, cervicovaginal and anorectal microbial swabs were obtained from 35 postmenopausal women (10 controls, 14 EAC and 11 USC) undergoing hysterectomy. Extracted DNA was PCR amplified using barcoded 16S rRNA gene V4 primers. Sequenced libraries were processed using QIIME2. Phyloseq was used to calculate α- and β- diversity measures. Biomarkers associated with case status were identified using ANCOM after adjustment for patient age, race and BMI. PICRUSt was used to identify microbial pathways associated with case status.

### Results

Beta-diversity of microbial communities across each niche was significantly different (R2 = 0.25, p < 0.001). Alpha-diversity of the uterine microbiome was reduced in USC (Chao1, p = 0.004 and Fisher, p = 0.007) compared to EAC. Biomarkers from the three anatomical sites allowed samples to be clustered into two distinct clades that distinguished controls from USC cases (p = 0.042). The USC group was defined by 13 bacterial taxa across the three sites (W-stat>10, FDR<0.05) including depletion of cervicovaginal *Lactobacillus* and elevation of

**Data Availability Statement:** Raw sequence fastq files used for this study have been uploaded to the sequence read archive under the NCBI BioProject

ID: PRJNA758386. A link to this data can be found at: https://urldefense.proofpoint.com/v2/url?u=http-3A__www.ncbi.nlm.nih.gov_bioproject_758386&d=DwICaQ&c=slrrB7dE8n7gBJbeO0g-IQ&r=bZTdvuHZchg_MRc9WzfTVA&m=

uterine *Pseudomonas*. PICRUSTt analysis revealed highly significant differences between the USC-associated clades within the cervicovaginal and uterine microbiota.

## Conclusions

The microbial diversity of anatomic niches in postmenopausal women with EAC and USC is different compared to controls. Multiple bacteria are associated with USC case status including elevated levels of cervicovaginal *Lactobacillus*, depletion of uterine *Pseudomonas*, and substantially different functional potentials identified within cervicovaginal and uterine niches.

## Introduction

Within the human body, there exists multitudes of microorganisms which colonize mucosal surfaces to form distinct microbial communities within body site niches. Together, these microorganisms outnumber human cells by a factor of 10 and are collectively termed the *microbiome* [1]. Bacterial communities existing in anatomic niches such as the gastrointestinal tract, oral cavity and genitourinary system can be classified as mutualists (symbiotically effective microbes), commensals (microbes that are neither harmful nor helpful to the host) and pathogens (colonizing microbes which may cause potential harm to the host) [2]. Mechanisms by which microbiota exert their influences on human health are not well-defined, but under certain circumstances the relative abundance of certain bacterial communities can become altered, thereby disrupting normal homeostasis and resulting in human disease [3]. Microbial disequilibrium can result in endothelial dysfunction, alteration of host immune systems, bacterial translocation, chronic inflammation, and global genomic instability, all of which are hallmarks of cancer [4]. Indeed multiple studies have linked alteration of the normal micriobiome with development of human cancers [5–9].

While disruption of the genitourinary microbiome may potentially promote gynecologic carcinogenesis, the exact role of the microbiome in gynecologic cancers remains unclear [10]. The vagina is a complicated microbial niche which allows for survival and proliferation of a number of both beneficial bacteria as well as opportunistic pathogens. The "ascending infection theory" postulates that disruption in the vaginal microbiome can allow pathogenic bacteria to ascend into the upper genital tract and cause polymicrobial infections, resulting in inflammation of the uterus, fallopian tubes and ovaries, a condition known as pelvic inflammatory disease (PID) [11]. As chronic inflammation is an etiologic factor in most cancers, this establishes a plausible hypothesis for how microbial dysregulation may promote tumor formation [12]. Furthermore, endometrial cancer is promoted by obesity, hormonal imbalances, diabetes and metabolic syndrome, all of which may promote changes in the microbiome [13–16]. The objective of the present study was to characterize the endometrial, cervicovaginal and anorectal microbiota of postmenopausal women undergoing hysterectomy for endometrioid and serous uterine cancers relative to controls undergoing hysterectomy for benign conditions.

## Materials and methods

### Subject recruitment

This prospective study was approved by the Albert Einstein College of Medicine Institutional Review Board and the Protocol Review and Monitoring Committee. Eligible subjects

scheduled for hysterectomy were recruited from a gynecologic oncology clinic from September 2016 thru April 2019. Patients meeting inclusion criteria were approached initially by a gynecologic oncologist; those expressing interest in participation then met with a clinical study coordinator and were given the option of enrollment in the study and those willing provided informed written consent. Women were considered eligible for inclusion if they were English or Spanish-speaking, post-menopausal (defined as 50 years of age or older who have not had menses for 12 consecutive months or more) with biopsy proven well-differentiated endometrioid endometrial adenocarcinoma (EAC), uterine serous carcinoma (USC) or with non-cancerous conditions requiring hysterectomy. Exclusion criteria included history of prior cancer, prior chemotherapy or radiation therapy, prior bariatric surgery, history of human immunodeficiency virus or PID, use of douching, hormone therapy, systemic or local antibiotics, pro-biotics or anti-fungal medications within 2 weeks of initial consultation, or confirmed urinary tract or vaginal infection such as bacterial vaginosis, sexually transmitted disease or candidiasis within 1 month of initial consultation.

A total of 55 potentially eligible patients were evaluated during the study period. Of these, 11 women had unexpected final pathology results rendering them ineligible, 8 lacked specimen collection and 1 was determined after enrollment to be pre-menopausal. These women were excluded from the study.

## Sampling procedures

After induction of anesthesia and prior to antibiotic administration and vaginal preparation, the patient was placed in dorsal lithotomy position. A sterile bi-valve speculum was inserted inside the vagina to expose the uterine cervix and vaginal canal. A sterile q-tip applicator was used to swab the vaginal fornices and ectocervix and was then placed into a 1 mL tube of Specimen Transport Medium (STM) (Digene Female Swab Specimen Collection Kit, Qiagen, CA). Anorectal swabs were obtained by inserting a nylon q-tip applicator approximately 1 cm beyond the anal verge and then placed in a tube containing STM. The patient was then surgically prepped and the hysterectomy proceeded as planned. After removal of the uterus, the specimen was placed on a sterile field and opened coronally using a sterile scalpel in order to expose the endometrial cavity. Swabs were obtained from the endometrial cavity and placed in STM as described above. The surgical specimen was then handed over to pathology for routine clinical diagnostic testing. All research samples were stored in a -20˚ freezer within one hour of collection.

## Clinicopathologic data

After pathologic confirmation of hysterectomy specimens as containing either non-cancerous tissue, well-differentiated endometrioid carcinoma or uterine serous cancer, clinical and pathologic information was abstracted from the medical record and stored in a secure database. Collected variables included patient age, race, ethnicity, body mass index (BMI in kg/m$^2$), parity, chart-indicated diagnosis of medical comorbidities (diabetes, hypertension, cardiovascular disease, smoking status), stage of cancer, tumor size, and presence of lymphovascular space invasion. Continuous variables were reported as means ± standard deviations. Categorical data were presented as number of patients with percentages. Bivariate analysis was performed to assess differences of covariates between groups (benign, endometrioid and serous). Continuous variables were assessed using one way analysis of variance (ANOVA) or Kruskal-Wallis tests as appropriate, whereas categorical and dichotomous variables were examined using $x^2$ or Fisher exact tests as appropriate. Data analysis was performed using Stata 14.2.

### DNA extraction and next-generation sequencing

DNA from samples stored in STM was extracted using the QIAamp DNA Kit (Qiagen, CA) following their standard protocol, modified by pre-incubation with Proteinase K and agitation with glass beads. Sterility was maintained by processing these samples in a sterile Biosafety Cabinet in an isolated extraction room. An aliquot of DNA from each STM sample was PCR amplified using barcoded primers annealing to conserved sequences within the V4 region (~250 base pairs) of the 16S ribosomal RNA (rRNA) gene. The 16SV4 rRNA region was chosen because of it's robust ability to effectively distinguish between vaginal bacterial species when compared to alternate 16S regions [17]. Barcoded and purified DNA libraries were sequenced using an Illumina MiSeq (Illumina Inc., San Diego, CA) using paired-end reads.

### Bioinformatics analysis

Paired end Illumina reads were left trimmed to remove bases that had a PHREAD quality score of 25 or lower using prinseq-lite [18]. Quality controlled reads were then demultiplexed using dual Golay barcodes [19] using denovobarcode [20]. Reads were then merged using PANDAseq [21] and processed using the QIIME2 platform [22]. Briefly, VSEARCH [23] was used to cluster sequences into operational taxonomic units (OTUs) and to assign taxonomy using the Greengenes [24] database. The *phyloseq* [25] package was used to import data into R [26] for final processing. Alpha and beta diversity was calculated using *phyloseq* [25] and *ggplot2* [27] was used to make final figures. ANCOM [28] was used to identify bacterial markers for endometrial cancer with adjustment for multiple testing (FDR<0.05) as well as adjustment for age and race. Ward.D2 [29] algorithm was used to perform hierarchical clustering using identified biomarkers. ANCOM was utilized in order to overcome the compositional limitations inherent to the study of the microbiome. For more details on this issue and how ANCOM addresses it see the cited reference [30].

## Results

A total of 35 patients had microbiome specimens collected and analyzed (14 with well-differentiated EAC, 11 with USC, and 10 controls) (Table 1). The median age of the final cohort was 63.6 ± 9.2 years with no significant differences between groups (p = 0.47). Most patients were obese with a median BMI of 33.6 (IQR 26.3, 38.8) and there were no significant differences in BMI between groups (p = 0.10). There were no significant differences in race, ethnicity, medical comorbidities (diabetes, hypertension, hyperlipidemia) or smoking status across histologic groups. Of the 25 women with cancer, most (88.0%) had early stage (I or II) cancer with no significant differences between groups in terms of extent of disease (p = 0.57), tumor size (p = 0.68), or presence of lymphovascular space invasion (p = 0.13). Of the control women, 6 (60%) had uterine fibroids, 2 (20%) had ovarian cystadenomata, 1 (10%) had adenomyosis and 1 (10%) had pelvic organ prolapse.

The top 20 bacterial genera within the cervicovaginal, uterine and anorectal microbiota revealed significant community separation based on anatomical site (**Fig 1**). The cervicovaginal microbiome was found to be dominated by *Lactobacillus* with additional elevated abundances of *Prevotella* and *Gardnerella* relative to other genera. There was a clear separation of the uterine microbiome from the other two niches and this body site was the most even in terms of genus diversity with a dominance of *Flavobacterium*. The anorectal microbiome was dominated by either *Prevotella* or *Bacteroides* (depending on the individual sample). PERMANOVA analysis using weighted unifrac distances revealed significant separation between the three anatomical niches (**Fig 2**, R2 = 0.25, p < 0.001).

**Table 1. Characteristics of subjects (N = 35) included in the study.**

| Characteristics | Total Cohort | Benign | Endometrioid | Serous | P Value |
|---|---|---|---|---|---|
| | (N = 35)* | (N = 10) | (N = 14) | (N = 11) | |
| Age (years) | 63.6 ± 9.2 | 59.0 ± 7.2 | 61.4 ± 9.4 | 70.5 ± 6.6 | 0.47 |
| Body mass index (kg/m$^2$) | 33.6 (26.4, 38.8) | 29.2 (26.4, 36.3) | 38.0 (33.1, 42.5) | 32.4 (24.7, 36.1) | 0.10 |
| Parity | 3 (1, 4) | 3 (3, 4) | 2 (1, 3) | 3 (1, 4) | 0.15 |
| Race | | | | | |
| White | 20 (57.1) | 3 (30.0) | 11 (78.6) | 6 (54.6) | 0.07 |
| Non-White | 15 (42.9) | 7 (70.0) | 3 (21.4) | 5 (45.5) | |
| Ethnicity | | | | | |
| Hispanic | 21 (60.0) | 8 (80.0) | 7 (50.0) | 6 (54.6) | 0.38 |
| Non-Hispanic | 14 (40.0) | 2 (20.0) | 7 (50.0) | 5 (45.4) | |
| Medical co-morbidities | | | | | |
| Diabetes | 10 (28.6) | 4 (40.0) | 5 (35.7) | 1 (9.1) | 0.21 |
| Hypertension | 28 (80.0) | 8 (80.0) | 9 (64.3) | 11 (100.0) | 0.08 |
| Hyperlipidemia | 18 (51.4) | 7 (70.0) | 5 (35.7) | 6 (54.6) | 0.28 |
| Smoking status | | | | | |
| Never or former smoker | 32 (91.4) | 8 (80.0) | 13 (92.8) | 11 (100.0) | 0.36 |
| Current everyday smoker | 3 (8.6) | 2 (20.0) | 1 (7.1) | 0 (0) | |
| Cancer Stage[†] | | | | | |
| Stage I / II | 22 (88.0) | - | 13 (92.9) | 9 (81.8) | 0.57 |
| Stage III / IV | 3 (12.0) | - | 1 (7.1) | 2 (18.2) | |
| Tumor size (cm) [†] | 3.5 (2.4, 5.0) | - | 3.3 (1.7, 5.2) | 3.9 (2.4, 5.0) | 0.68 |
| Lymphovascular space invasion[†] | 5 (20.0) | - | 1 (7.1) | 4 (36.4) | 0.13 |

* Data with plus-minus values represent means ± standard deviation, otherwise reported as median (interquartile range). Categorical data are presented as N (%) associated with odds ratios and 95% confidence intervals.

[†] Analysis based only on the 25 study subjects with cancer.

Microbial alpha diversity analyses demonstrated that serous cancers were characterized by a significant reduction of diversity within taxa of the control uterine microbiome based on the Chao1 (p = 0.004) and Fisher (p = 0.007) measures with a reduction in Shannon diversity (p = 0.094) (**Fig 3**). The anorectal and cervicovaginal microbiome alpha diversity was not correlated with case status with the exception of Chao1, which was reduced in endometrioid cancers (p = 0.026).

ANCOM analysis was performed to identify biomarkers that can distinguish patients across the three groups following adjustment for patient age, race and BMI. **Fig 4** shows composite hierarchical clustering performed using all biomarkers identified in the ANCOM analysis where W-stat>10 and FDR<0.05. Two distinct microbiome clusters were observed with cluster indicated as "Biomarker Cluster 1" having a significantly higher prevalence of serous cancer patients (7/10) as compared to "Biomarker Cluster 2" where the majority of the benign cases are found (n = 6/7) (p = 0.042). No significant differences were found when comparing the presence of benign cases vs. endometroid (p = 0.30) or endometroid vs. serous cases (p = 0.68). Biomarker cluster 1 had several bacterial genera elevated including uterine *Pseudomonas* while Biomarker cluster 2 was distinguished by a dominance of cervicovaginal *Lactobacillus* and *Clostridium* genera.

In order to identify potentially relevant functional pathways that may be associated with the serous cancer associated microbiome clusters, PICRUSt analysis was performed at KEGG level 3. **Fig 5A** shows PCA within the cervicovaginal region. A total of 49 pathways were identified

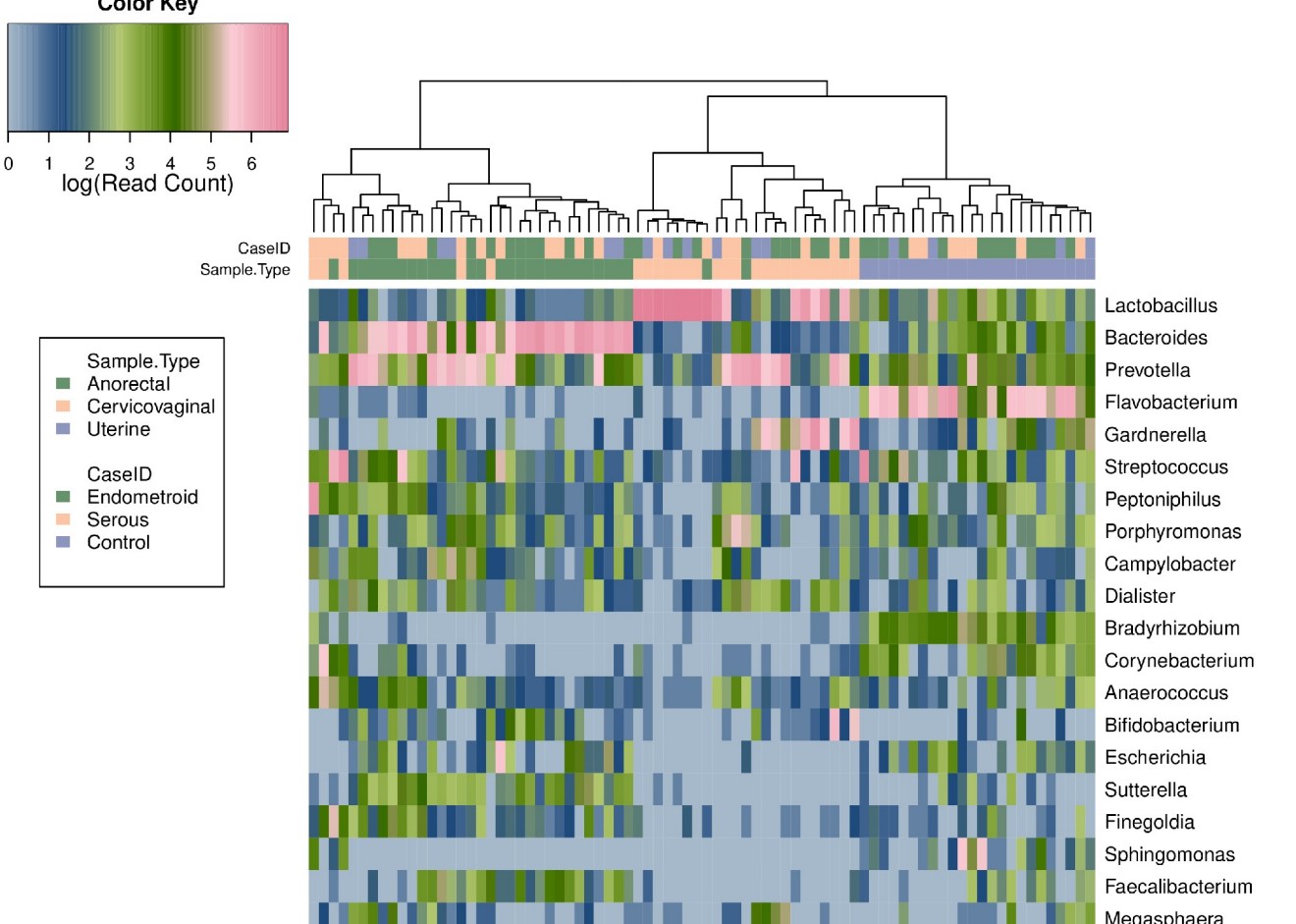

**Fig 1. Heatmap of 16S V4 reads from uterine, cervicovaginal and anorectal samples.** The 20 most abundant genera of the uterine, cervicovaginal and anorectal microbiota are shown and abundance is represented by color based on a log scale shown at the top left. The names of the bacterial genera are shown at the right of the heatmap. There is a clear separation of the uterine microbiome from the other two niches. The cervicovaginal and anorectal microbiota separate out with the exception of 6 samples.

to be significant (**S1 Fig**). In order to present the data in a concise manner, only pathways with a corrected FDR<0.05 and a fold change of 3 or greater are shown in **Fig 5B**. Similarly **Fig 5C** shows the PCA performed for the uterine samples with significantly differentially abundant pathways after correction for multiple testing (FDR<0.05) in **Fig 5D**.

## Discussion

Our study demonstrates significant differences in bacterial diversity within the uterine, cervicovaginal and anorectal microbiota of women with EAC, USC, and non-malignant controls suggesting a potential role of disruption of the normal microbiome in these histologic types of uterine cancer. Walther-António and colleagues performed a similar study examining microbial differences in women with benign conditions, endometrial hyperplasia and endometrial cancer and demonstrated that endometrial cancers were enriched for Firmicutes, Spriochaetes, Actinobacteria, and Proteobacteria [31]. They also found an association between *Atopobium vaginae* and *Porphyromonas* along with a high vaginal pH and the presence of endometrial cancer. Although their study did contain 3 patients with serous cancer, their analysis did not

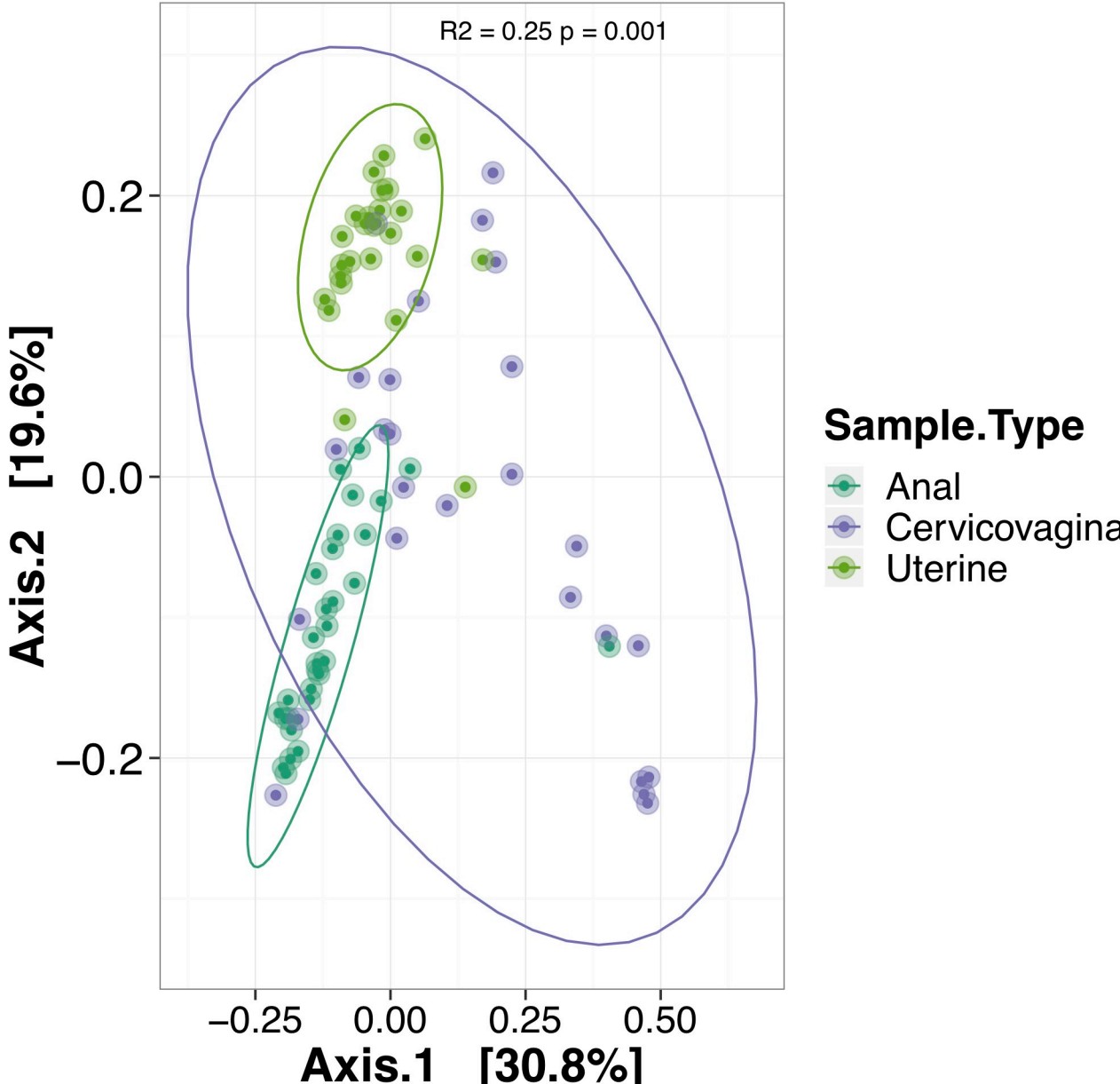

**Fig 2. PCoA performed across sampled anatomical sites.** The PCoA performed using weighted unifrac distances after subsampling each sample library to 1,000 reads. Ellipses represent 95% confidence intervals for site based sample clustering. PERMANOVA analysis indicates significant community separation at the genus level based on anatomic location (R2 = 0.25, p < 0.001).

differentiate between EAC and more aggressive uterine histologies. Our study demonstrates significant reduction in microbial diversity of USC specimens relative to EAC specimens or controls.

Our study is also important in that it provides a characterization of a postmenopausal population. Until the 1980s, it was thought that a healthy uterus was sterile [32]. The endometrial cavity is a body niche with low bacterial abundance with 100–10,000 fold fewer

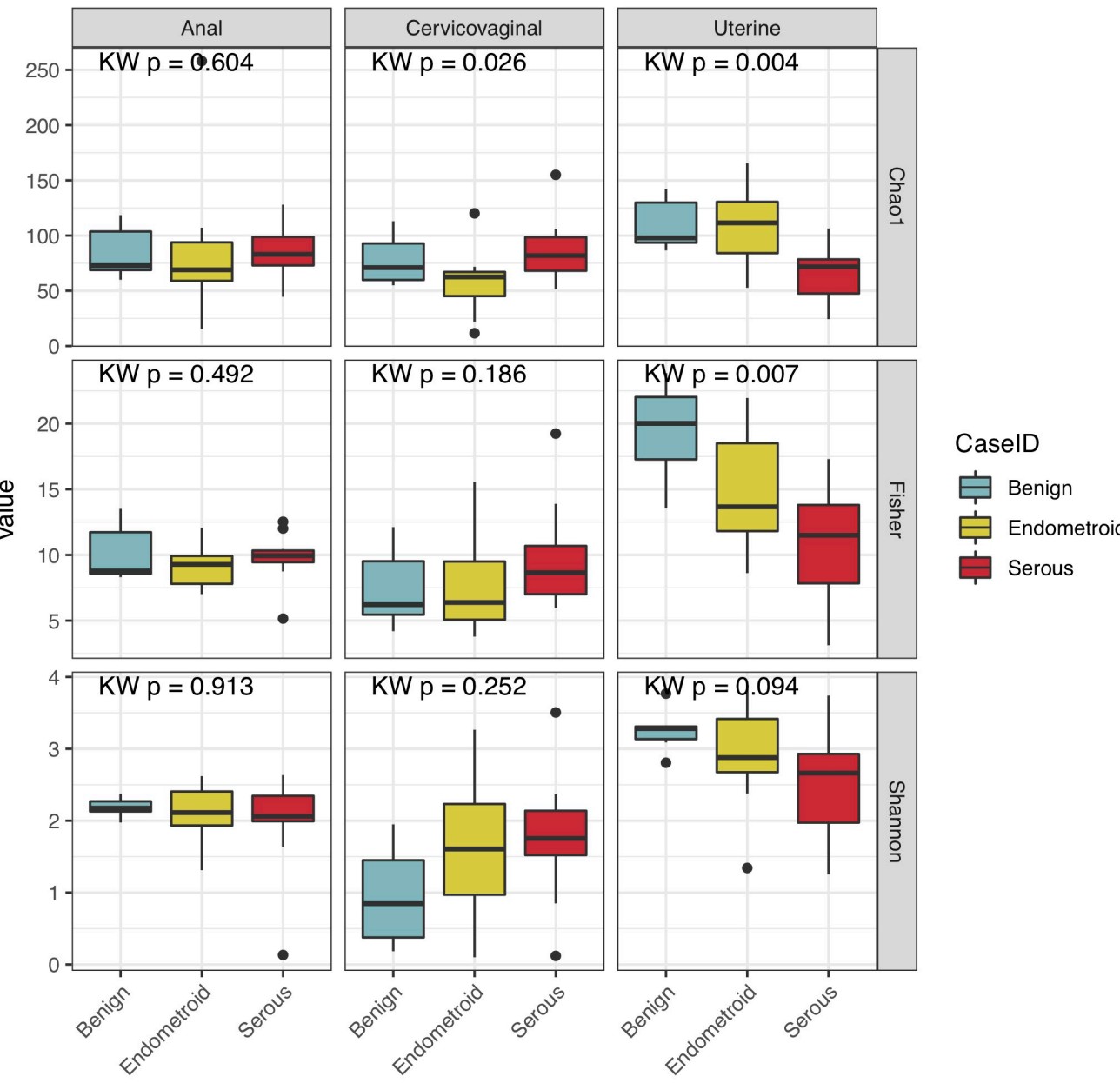

**Fig 3. Microbial alpha diversity analysis of anorectal, cervicovaginal and uterine samples.** The plot indicates that serous cancer is characterized by a significant reduction of the uterine microbiome diversity based on the Chao1 (p = 0.004) and Fisher (p = 0.007) measures with a marginally significant reduction in Shannon Diversity (p = 0.094). The Kruskal-Wallis (KW) statistic is shown at the top of each plot.

bacteria compared with the vagina [33]. The bacterial paucity of the endometrium compared with difficulty culturing most uterine bacteria has made it difficult to characterize the uterine microbiome until the advent of 16S rRNA next generation sequencing technologies. Nevertheless, the majority of extant literature regarding the endometrial microbiome focuses on pre-menopausal women and the association between microbiome dysregulation and adverse pregnancy outcomes [34–37]. Our study shows that the uterine microbiome of

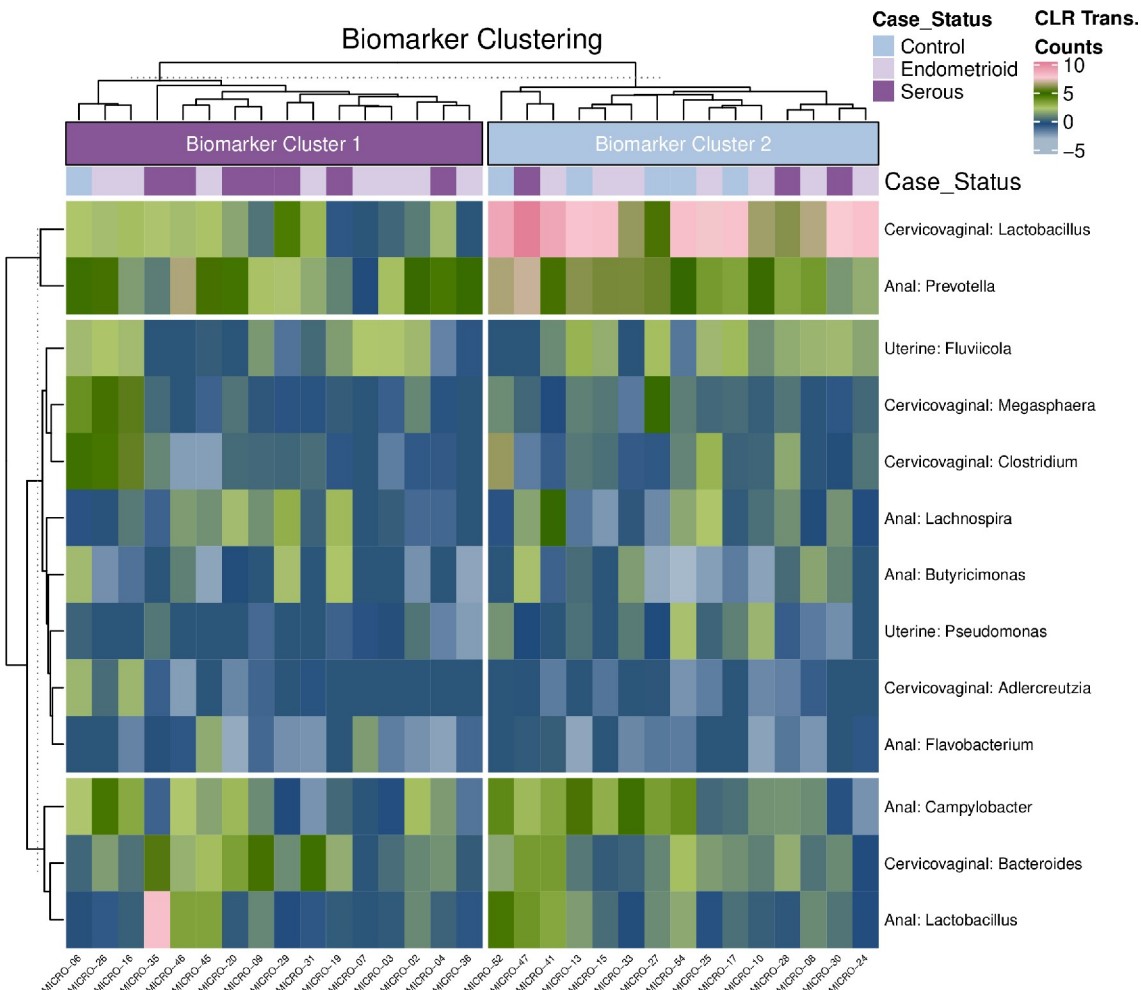

**Fig 4. Composite heatmap of ANCOM identified biomarkers.** Heatmap shows all ANCOM biomarkers with a W-stat>10 and FDR<0.05 clustered using the ward.D2 algorithm. Two microbial clades can be observed following k-means clustering based on the resulting dendrogram. Cluster designated as "Biomarker Cluster 1" has a significantly higher proportion of serous cancer cases (7/10) as compared to "Biomarker Cluster 2" where the majority of the benign cases are grouping (6/7) (p = 0.042). No significant separation is occurring with respect to the identified clusters and benign vs. endometroid samples (p = 0.30) or the endometroid vs. serous samples (p = 0.68). "Biomarker Cluster 1" has a significant depletion of cervicovaginal *Lactobacillus* and *Clostridium* and a higher overall abundance of uterine *Pseudomonas*.

postmenopausal women was comparable across histologies in terms of genus diversity with a dominance of *Flavobacterium*. Franasiak et al. investigated the uterine microbiome at the time of IVF and embryo transfer and also found *Flavobacterium* to be an abundant taxa of the uterine microbiome [38]. Walsh et al also examined the endometrial microbiome composition in patients with and without endometrial cancer and identified significantly increased alpha diversity amongst post-menopausal patients [39]. In their study, the post-menopausal endometrium demonstrated enrichment of *Anaerococcus*, *Peptoniphilus* and *Porphyromonas*, but only 7 uterine specimens were examined of post-menopausal women. All 35 women in our study were post-menopausal and each of them had sufficient endometrial samples available for analysis.

In addition to a reduction of uterine bacterial diversity, we were also able to demonstrate a significant correlation between lower vaginal *Lactobacillus* and elevated uterine

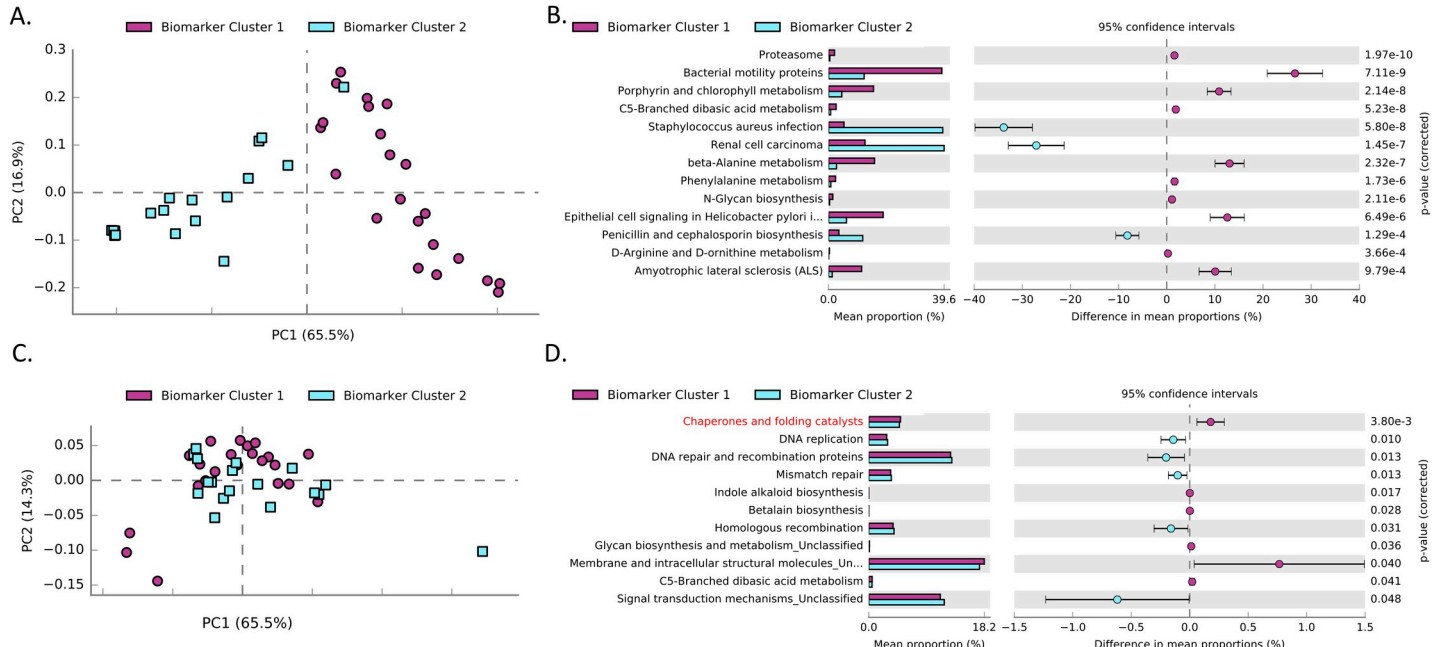

**Fig 5. Functional potential of uterine serous cancer associated microbiome clusters.** Panel A shows PCA performed using the KEGG level 3 for bacteria present within the cervicovaginal samples. Full sample separation can be observed across the principle component (PC1, 65.5%) between the two-microbiome clusters. Panel B shows significantly differentially abundant pathways within the cervicovaginal samples that have a minimum fold ratio of 3 or greater between the two clusters. Panel C shows the PCA performed using the uterine samples. Panel D shows significantly differentially abundant pathways within the uterine samples with the "Chaperone and folding catalyst" pathway highlighted in red.

*Pseudomonas* associated with USC case status. Vaginal *Lactobacillus* has been previously associated with several gynecological cancers such as cervical [40], ovarian [41] and endometrial [31] cancers as well as general health of the cervicovaginal tract [42]. *Pseudomonas* has also been recently implicated in association with endometrial cancer by Winters [43] as well as Walther-António [31] and colleagues using 16S rRNA amplicon sequencing of uterine samples. *Lactobacillus* may be acting to limit carcinogenesis by reducing local inflammation through cytokine modulation. Other studies have demonstrated a positive association between vaginal *Lactobacillus* and genitourinary health [44, 45]. Furthermore we have integrated biomarkers from across several sampled regions to show that we can effectively separate Serous cancers from controls using unsupervised clustering. This may suggest that the use of disease related biomarkers from across multiple anatomical regions may provide more clinically relevant groupings than the more broad CST categories. Additionally, we identified several pathways previously associated with several cancer subtypes [46]. Particularly interesting may be our identification of the elevation of the chaperone and folding pathway within the uterine microbiome within the USC associated microbial biomarker clade. Overexpression of the HSF1 protein in particular has been linked with significantly lower survival in endometrial cancer patients [47] and given our finding, this may have a correlation to the composition of the urogenital microbiome.

Strengths of our study include the fact that we have characterized microbial composition of anogenital body sites in a racially and ethnically diverse population of postmenopausal women with uterine cancer and contrasted these findings with a control group of postmenopausal women without cancer. Although our cohort is small, it is among the largest published reports characterizing the uterine microbiome in postmenopausal women. Our histologic groups were well balanced in terms of their clinical and pathologic covariates. We also excluded enrollment

of premenopausal women and those with recent infections or use of probiotics and antibiotics in order to reduce confounding of our results. It can therefore be assumed that the differences seen between groups can be attributed to histologic differences rather than lifestyle factors that would be difficult to control for in our analyses. It should also be noted that the functional pathways analysis, although shown to be highly correlated with true functional composition of sampled communities [48], is an estimate based on the 16S rRNA sequencing and should be verified using shotgun metagenomic approaches.

Nevertheless there are limitations of our study. Although we made all attempts to collect specimens in a sterile fashion, sterility can never be completely ensured. It is difficult to obtain samples from the endometrial cavity without passing through the cervical os or bivalving the uterus from the side as we have done in this study. Furthermore, the use of a uterine manipulator during hysterectomy may influence the results. Additional prospective studies are required to longitudinally sample the microbiota of postmenopausal women and determine if progressive disruption of the microbiome contributes to endometrial carcinogenesis. Moreover prospective clinical trials will help determine if interventions such as probiotic administration may mitigate risk of gynecologic malignancy.

## Conclusion

The microbial diversity of anatomical ecological niches in postmenopausal women with EAC and USC is different compared to benign controls. This difference is both in terms of community structure as defined by a reduction of microbial alpha diversity within the uterus and by differences of bacterial taxa within the cervicovaginal and uterine regions. Microbial composition and presence of specific functional pathways may be necessary for development of endometrial cancer and further investigation using prospective datasets is warranted.

## Supporting information

**S1 Fig. 49 significantly differentially abundant pathways (KEGG level 3) within the serous cancer associated microbiome clusters imputed using PICRUSt.**
(TIF)

## Author Contributions

**Conceptualization:** Gregory M. Gressel, Mykhaylo Usyk, Marina Frimer, Robert D. Burk.

**Data curation:** Gregory M. Gressel, Mykhaylo Usyk, Robert D. Burk.

**Formal analysis:** Gregory M. Gressel, Mykhaylo Usyk, Robert D. Burk.

**Investigation:** Gregory M. Gressel, Mykhaylo Usyk, Robert D. Burk.

**Methodology:** Gregory M. Gressel, Mykhaylo Usyk, Robert D. Burk.

**Project administration:** Robert D. Burk.

**Resources:** Robert D. Burk.

**Supervision:** D. Y. S. Kuo, Robert D. Burk.

**Writing – original draft:** Gregory M. Gressel.

**Writing – review & editing:** Gregory M. Gressel, Mykhaylo Usyk, Marina Frimer, D. Y. S. Kuo, Robert D. Burk.

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
