## [Decision Letter · Decision Letter 0]

22 Jun 2021

PONE-D-21-05233

Characterization of the endometrial, cervicovaginal and anorectal microbiota in post-menopausal women with endometrioid and serous endometrial cancers

PLOS ONE

Dear Dr. Gressel,

Thank you for submitting your manuscript to PLOS ONE. After careful consideration, we feel that it has merit but does not fully meet PLOS ONE’s publication criteria as it currently stands. Therefore, we invite you to submit a revised version of the manuscript that addresses the points raised during the review process.

Thank you for your patience during this unusually long review period, due to the pandemic, it has been difficult to find enough available reviewers and acquire timely reviews.  As such, I was only able to secure one set of reviews for your manuscript.  However, I agree with the reviewer's assessment of the submission, and with their recommendations to improve the manuscript.

We look forward to receiving your revised manuscript.

Kind regards,

Suzanne L. Ishaq, PhD

Academic Editor

PLOS ONE

Journal Requirements:

Reviewers' comments:

Reviewer's Responses to Questions

**Comments to the Author**

1. Is the manuscript technically sound, and do the data support the conclusions?

Reviewer #1: Yes

2. Has the statistical analysis been performed appropriately and rigorously? 

Reviewer #1: Yes

3. Have the authors made all data underlying the findings in their manuscript fully available?

Reviewer #1: No

4. Is the manuscript presented in an intelligible fashion and written in standard English?

Reviewer #1: Yes

5. Review Comments to the Author

Reviewer #1: This study aimed to discover if the microbiome at three body sites is different among post-menopausal women undergoing hysterectomy. They swabbed these three sites, which then underwent 16S rRNA sequencing on the V4 region. QIIME2 was used for taxonomic identification at the genus level, and PICRUST was used for functional annotation.

Overall, this study found differences in alpha diversity at these three body sites across three clinical groups (endometrial cancer, uterine serous cancer, and other benign conditions). They also discovered potential metabolic pathways associated with these different clinical groups that shotgun metagenomics can confirm. Authors do a good job of stating their findings (without overstating) and tying them to biological mechanisms and clinical importance.

This study is unique in that it focuses on post-menopausal women, whereas much of the literature on endometrial microbiome focuses on those of reproductive age.

Page 9 line 58: use of the word dysbiosis: dysbiosis is a vague word that means a lot of different things depending on who you ask. It seems as though the phrase “the microbiome” could easily replace dysbiosis with the same meaning. I suggest either replacing the phrase “dysbiosis” or defining its use in the statement of translational relevance.

Page 10 line 78, introduction: manuscript citations 5 through 9 are studies linking “dysbiosis” to human cancer, however, upon further inspection, these citations appear to link specific taxa to the development of cancer. I don’t think these citations fit that manuscript’s definition of “dysbiosis” so I would change the word “dysbiosis” in this line.

Page 10 line 79 states “some authors postulate”; it is my understanding that we “should” be stating findings as “the research suggests/has evidence for”, etc, so as to not “call out” other authors in a manuscript.

Citation 11 on page 10 line 84: “vaginal dysbiosis has been link[ed] to bacterial vaginosis” This citation discusses PID, not BV. I’m assuming that this is the wrong citation here? I would also be careful about what citation you use, because the diverse community state type is associated with bacterial vaginosis, but is NOT itself a dysbiotic state.

Authors appear to use “dysbiosis”, “microbial disequilibrium”, and “microbial dysregulation” interchangeably. If all three terms are used to refer to the same phenomena, this needs to be explicitly stated. I would lean toward removing “dysbiosis” from the manuscript in favor of the other two terms due to the baggage that comes with the term “dysbiosis”

Page 12 line 151: it would be great to include rationale for why the V4 region was selected for 16S rRNA sequencing

Citation 27: page 12 line 163: I read the ANCOM pub just now, and this seems like a very interesting and statistically sound approach. It’s not clear to me from the manuscript if this is a program (like QIIME), a package (like phyloseq), or just a method that one implements themselves.

Page 13 line 193: a p value greater than 0.05 is not marginally significant; 0.45 would be marginally significant. I would change this language prior to publication.

Page 14 lines 201-203: It’s not clear to me what the significance of the “biological cluster [1|2]” is, and this doesn’t appear to be further discussed in the discussion section. I would either remove this, include it in the discussion/state it’s relevance, or, if I missed this within the manuscript, state it more clearly.

Page 15 lines 245-246: I don't remember reading in the methods/results that depleted vaginal Lactobacillus >> elevated uterine Pseudomonas in those with USC. I like the discussion of this (lots of lit, proposed biological mechanisms with clinical links), but don’t remember this in methods. Is this related to the “biological clustering”? If so, this isn’t currently clear in the manuscript. Should be more explicit in the manuscript prior to publication

I like your discussion and justifications in lines 265-271

I appreciate that there is not a discussion of community state types - it would be hard to draw conclusions of CSTs with the V4 region alone and in general, the CST does not provide clinically relevant information. I am explicitly stating this since there is often pressure to include CSTs in analysis despite the reasons stated here.

Some figures (fig 1, 4) may be hard to read for those with red-green colorblindness, it is worth adjusting the color palette before publication since this is a common condition.

Manuscript does not state where the data (fastq/fasta files) are available.

6. PLOS authors have the option to publish the peer review history of their article (what does this mean?). If published, this will include your full peer review and any attached files.

Reviewer #1: **Yes: **Emily F. Wissel

---

## [Author Response · Author response to Decision Letter 0]

7 Sep 2021

Journal Requirements:

***We confirm that we have met the style requirements outlined in these documents.

Comments to the Author:

1. Is the manuscript technically sound, and do the data support the conclusions?

Reviewer #1: Yes

2. Has the statistical analysis been performed appropriately and rigorously?

Reviewer #1: Yes

3. Have the authors made all data underlying the findings in their manuscript fully available?

Reviewer #1: No

***We have added the 16S V4 amplicon sequencing data to SRA as described below.

4. Is the manuscript presented in an intelligible fashion and written in standard English?

Reviewer #1: Yes

***Thank you for reviewing our manuscript and for your insightful comments. We have done our best to address each of the comments in our revised manuscript.

Review Comments to the Author

Reviewer #1: This study aimed to discover if the microbiome at three body sites is different among post-menopausal women undergoing hysterectomy. They swabbed these three sites, which then underwent 16S rRNA sequencing on the V4 region. QIIME2 was used for taxonomic identification at the genus level, and PICRUST was used for functional annotation.

Overall, this study found differences in alpha diversity at these three body sites across three clinical groups (endometrial cancer, uterine serous cancer, and other benign conditions). They also discovered potential metabolic pathways associated with these different clinical groups that shotgun metagenomics can confirm. Authors do a good job of stating their findings (without overstating) and tying them to biological mechanisms and clinical importance.

This study is unique in that it focuses on post-menopausal women, whereas much of the literature on endometrial microbiome focuses on those of reproductive age.

Page 9 line 58: use of the word dysbiosis: dysbiosis is a vague word that means a lot of different things depending on who you ask. It seems as though the phrase “the microbiome” could easily replace dysbiosis with the same meaning. I suggest either replacing the phrase “dysbiosis” or defining its use in the statement of translational relevance.

***Thank you for drawing attention to the semantic differences in the literature surrounding this term “dysbiosis.” While there is certainly precedent for this term in the literature regarding the gynecologic microbiome, you are correct that the term is not often well-defined or consistent across studies. We initially had planned to define this term in the “statement of translational relevance,” in light of your other comments below, we have replaced this term with “disruption in the normal gynecologic microbiome” which is our intended meaning in using the word.

Page 10 line 78, introduction: manuscript citations 5 through 9 are studies linking “dysbiosis” to human cancer, however, upon further inspection, these citations appear to link specific taxa to the development of cancer. I don’t think these citations fit that manuscript’s definition of “dysbiosis” so I would change the word “dysbiosis” in this line.

***Thank you. We have replaced all uses of the term “dysbiosis” in favor of “disruption in the normal gynecologic microbiome.”

Page 10 line 79 states “some authors postulate”; it is my understanding that we “should” be stating findings as “the research suggests/has evidence for”, etc, so as to not “call out” other authors in a manuscript.

***Thank you for pointing this out. We have revised the manuscript to correct this verbiage. 

Citation 11 on page 10 line 84: “vaginal dysbiosis has been link[ed] to bacterial vaginosis” This citation discusses PID, not BV. I’m assuming that this is the wrong citation here? I would also be careful about what citation you use, because the diverse community state type is associated with bacterial vaginosis, but is NOT itself a dysbiotic state.

***We thank the reviewers for the opportunity to clarify this point. The cited reference: Wang Y, Zhang Y, Zhang Q, Chen H, Feng Y. Characterization of pelvic and cervical microbiotas from patients with pelvic inflammatory disease. J Med Microbiol 2018; refers to a study from Zhejiang university in which 38 patients with pelvic inflammatory disease and 19 control patients without pelvic inflammatory disease had 16s rRNA amplicon profiling to test the hypothesis that microbes in the vagina and cervix can spread to the upper genital tract and cause PID. You are absolutely correct that this reference does not refer to bacterial vaginosis and we have corrected the preceding statement as follows: “The vagina is a complicated microbial niche that allows for survival and proliferation of a number of both beneficial bacteria as well as opportunistic pathogens. The “ascending infection theory” postulates that disruption in the vaginal microbiome can allow pathogenic bacteria to ascend into the upper genital tract and cause polymicrobial infections, resulting in inflammation of the uterus, fallopian tubes and ovaries, a condition known as pelvic inflammatory disease.”

Authors appear to use “dysbiosis”, “microbial disequilibrium”, and “microbial dysregulation” interchangeably. If all three terms are used to refer to the same phenomena, this needs to be explicitly stated. I would lean toward removing “dysbiosis” from the manuscript in favor of the other two terms due to the baggage that comes with the term “dysbiosis”

***Thank you. We have replaced all uses of the term “dysbiosis” in favor of “disruption in the normal gynecologic microbiome.”

Page 12 line 151: it would be great to include rationale for why the V4 region was selected for 16S rRNA sequencing

***We thank the reviewer for the opportunity to clarify the use of 16S V4 rRNA sequencing in this project. The V4 region is the most commonly used region, which allows for more streamlined comparison’s across studies, and was recently demonstrate to show the best taxonomic resolution between vaginal flora (see PMID: 30535155, Willian Van Der Pol 2019). We have added justification along with the relevant citing text in the methods section:

***Lines 199-201: The 16SV4 rRNA gene region was chosen because of it’s robust ability to effectively distinguish between vaginal bacterial species when compared to alternate 16S regions. ANCOM is a method. In a simple explanation it just uses all combinations of microbial ratios to overcome compositional limitations of the microbiome when characterized using standard next generation sequencing. It does a few additional things like dealing with structural zeros within the data, but the testing of ratios is the core of this method. We have added an additional reference that delves into the use of microbial ratios as a means of dealing with compositionality that discusses ANCOM and several other similar approaches: (PMID 31222023, Morton 2019).

***Lines 214-216: ANCOM was utilized in order to overcome the compositional limitations inherent to the study of the microbiome. For more details on this issue and how ANCOM addresses it see: (PMID 31222023, Morton 2019).

Page 13 line 193: a p value greater than 0.05 is not marginally significant; 0.45 would be marginally significant. I would change this language prior to publication.

***We completely agree. We have removed the term “marginally significant”.

Page 14 lines 201-203: It’s not clear to me what the significance of the “biological cluster [1|2]” is, and this doesn’t appear to be further discussed in the discussion section. I would either remove this, include it in the discussion/state it’s relevance, or, if I missed this within the manuscript, state it more clearly.

***In Figure 4 we utilized the biomarkers identified with ANCOM from across the three sampling sites to perform unsupervised clustering. This resulted in the two distinct sample clusters that appeared to consistently segregate Serous cases from controls. The observation of these clusters is relevant as the use of disease relevant markers across several body sites may provide more clinically relevant groups as opposed to CST clusters that are often reported in the literature. We have added a statement regarding this in the discussion:

Lines 273-274: Furthermore we have integrated biomarkers from across several sampled regions to show that we can effectively separate Serous cancers from controls using unsupervised clustering. This may suggest that the use of disease related biomarkers from across multiple anatomical regions may provide more clinically relevant groupings than the more broad CST categories.

Page 15 lines 245-246: I don't remember reading in the methods/results that depleted vaginal Lactobacillus >> elevated uterine Pseudomonas in those with USC. I like the discussion of this (lots of lit, proposed biological mechanisms with clinical links), but don’t remember this in methods. Is this related to the “biological clustering”? If so, this isn’t currently clear in the manuscript. Should be more explicit in the manuscript prior to publication

***We thank the reviewer for the chance to clarify our results as these are important points within the discussion. The Lactobacillus/Pseudomonas relationship is presented in figure 4 and mentioned in the figure caption “ “Biomarker Cluster 1” has a significant reduction of cervicovaginal Lactobacillus and Clostridium and a higher overall abundance of uterine Pseudomonas. ” We have added additional text in the results section in order to make this more clear:

***Lines 223-225: Biomarker cluster 1 had several bacterial genera elevated including uterine Pseudomonas while Biomarker cluster 2 was distinguished by a dominance of cervicovaginal Lactobacillus and Clostridium genera.

I like your discussion and justifications in lines 265-271

***Thank you very much.

I appreciate that there is not a discussion of community state types - it would be hard to draw conclusions of CSTs with the V4 region alone and in general, the CST does not provide clinically relevant information. I am explicitly stating this since there is often pressure to include CSTs in analysis despite the reasons stated here.

***We thank the reviewer for this statement and agree that CSTs do not tend to offer clinically useful categories. This was in part what prompted us to use identified biomarkers to see whether we can generate clusters that are more directly relevant to endometrial cancer using the three sampling sites. We have added additional comments regarding this as stated in the response above. 

Some figures (fig 1, 4) may be hard to read for those with red-green colorblindness, it is worth adjusting the color palette before publication since this is a common condition.

Manuscript does not state where the data (fastq/fasta files) are available.

***We have uploaded the sequence fastqs to SRA (PRJNA758386) and added a link to the project in the new “Data Availability” section. We have also modified the figures to allow them to be interpreted by people with color blindness using the https://for-hue.herokuapp.com web site.

---

## [Decision Letter · Decision Letter 1]

15 Oct 2021

Characterization of the endometrial, cervicovaginal and anorectal microbiota in post-menopausal women with endometrioid and serous endometrial cancers

PONE-D-21-05233R1

Dear Dr. Gressel,

We’re pleased to inform you that your manuscript has been judged scientifically suitable for publication and will be formally accepted for publication once it meets all outstanding technical requirements.

Kind regards,

Suzanne L. Ishaq, PhD

Academic Editor

PLOS ONE

Additional Editor Comments (optional):

Reviewers' comments:

Reviewer's Responses to Questions

**Comments to the Author**

1. If the authors have adequately addressed your comments raised in a previous round of review and you feel that this manuscript is now acceptable for publication, you may indicate that here to bypass the “Comments to the Author” section, enter your conflict of interest statement in the “Confidential to Editor” section, and submit your "Accept" recommendation.

Reviewer #1: All comments have been addressed

2. Is the manuscript technically sound, and do the data support the conclusions?

Reviewer #1: Yes

3. Has the statistical analysis been performed appropriately and rigorously? 

Reviewer #1: Yes

4. Have the authors made all data underlying the findings in their manuscript fully available?

Reviewer #1: Yes

5. Is the manuscript presented in an intelligible fashion and written in standard English?

Reviewer #1: Yes

6. Review Comments to the Author

Reviewer #1: Authors addressed all comments fully. They added clarifications to the manuscript about why they selected the V4 region for 16S sequencing, clarified that "dysbiotic" referred to a disruption of the gynecological microbiome, clarified some of the background citations, further describes the biomarker clusters and their clinical relevance, and have made all their data publicly available.

This paper has novel findings about the microbiome at three body sites across three clinical conditions. These findings can pave they way for future clinical interventions and studies around these conditions through further evaluation of the identified biomarker clusters.

7. PLOS authors have the option to publish the peer review history of their article (what does this mean?). If published, this will include your full peer review and any attached files.

Reviewer #1: **Yes: **Emily F. Wissel

---

## [Editor Report · Acceptance letter]

20 Oct 2021

PONE-D-21-05233R1 

Characterization of the endometrial, cervicovaginal and anorectal microbiota in post-menopausal women with endometrioid and serous endometrial cancers 

Dear Dr. Gressel:

I'm pleased to inform you that your manuscript has been deemed suitable for publication in PLOS ONE. Congratulations! Your manuscript is now with our production department. 

Kind regards, 

on behalf of

Dr. Suzanne L. Ishaq 

Academic Editor

PLOS ONE